# Robust Handover Optimization Technique with Fuzzy Logic Controller for Beyond 5G Mobile Networks [note 1]

**DOI:** 10.3390/s22166199

**Published:** 2022-08-18

**Authors:** Saddam Alraih, Rosdiadee Nordin, Asma Abu-Samah, Ibraheem Shayea, Nor Fadzilah Abdullah, Abdulraqeb Alhammadi

**Affiliations:** 1Department of Electrical, Electronic and Systems Engineering, Faculty of Engineering and Built Environment, Universiti Kebangsaan Malaysia, Bangi 43600, Malaysia; 2Electronics and Communication Engineering Department, Istanbul Technical University, Istanbul 34467, Turkey or; 3Wireless Communication Centre, School of Electrical Engineering, Faculty of Engineering, Universiti Teknologi Malaysia, Johor Bahru 81310, Malaysia; 4Communication Systems and Networks Research Lab, Malaysia-Japan International Institute of Technology, University Teknologi Malaysia, Kuala Lumpur 54100, Malaysia

**Keywords:** 5G, B5G, 5G and beyond, self-optimization, MRO, HCP, mobility management, handover, small cells, robust handover technique

## Abstract

Mobility management is an essential process in mobile networks to ensure a high quality of service (QoS) for mobile user equipment (UE) during their movements. In fifth generation (5G) and beyond (B5G) mobile networks, mobility management becomes more critical due to several key factors, such as the use of Millimeter Wave (mmWave) and Terahertz, a higher number of deployed small cells, massive growth of connected devices, the requirements of a higher data rate, and the necessities for ultra-low latency with high reliability. Therefore, providing robust mobility techniques that enable seamless connections through the UE’s mobility has become critical and challenging. One of the crucial handover (HO) techniques is known as mobility robustness optimization (MRO), which mainly aims to adjust HO control parameters (HCPs) (time-to-trigger (TTT) and handover margin (HOM)). Although this function has been introduced in 4G and developed further in 5G, it must be more efficient with future mobile networks due to several key challenges, as previously illustrated. This paper proposes a Robust Handover Optimization Technique with a Fuzzy Logic Controller (RHOT-FLC). The proposed technique aims to automatically configure HCPs by exploiting the information on Reference Signal Received Power (RSRP), Reference Signal Received Quality (RSRQ), and UE velocity as input parameters for the proposed technique. The technique is validated through various mobility scenarios in B5G networks. Additionally, it is evaluated using a number of major HO performance metrics, such as HO probability (HOP), HO failure (HOF), HO ping-pong (HOPP), HO latency (HOL), and HO interruption time (HIT). The obtained results have also been compared with other competitive algorithms from the literature. The results show that RHOT-FLC has achieved considerably better performance than other techniques. Furthermore, the RHOT-FLC technique obtains up to 95% HOP reduction, 95.8% in HOF, 97% in HOPP, 94.7% in HOL, and 95% in HIT compared to the competitive algorithms. Overall, RHOT-FLC obtained a substantial improvement of up to 95.5% using the considered HO performance metrics.

## 1. Introduction

There has been enormous development in mobile cellular networks during the last decade. Fifth generation (5G) and beyond (B5G) technologies aim to provide revolutionary features, such as ultra-high speed, extremely low latency, huge capacity (wide bandwidth), a massive number of connected devices, all-time availability, coverage everywhere, low energy usage, and long battery life [1]. In addition to that, the B5G networks support connectivity under high mobility speeds.

To meet the increasing demand of user data throughput, the carrier frequency range has been further expanded, such as to Millimeter Wave (mmWave) frequencies in B5G cellular networks, and even Terahertz may work with the sixth generation (6G) and beyond networks. However, mmWave has limited coverage of 200 m. This means that several base stations (BSs) will cover a small area to improve the frequency reuse and the total network capacity. In addition, a larger number of BSs in a small area increase the number of handovers (HO) and mobility management challenges in B5G networks, especially for a user speed higher than 100 km/h.

Consequently, mobility management functions are essential and need continuous developments and investigations in mobile cellular networks. That involves various mobility management functions, such as the self-optimization for HO control parameters, handover decision algorithms, mobile routing algorithms, authentications, identification and tracking of changes in user locations connected to the cellular network. Furthermore, mobility management provides network connectivity to users at any location. Users can avail themselves of this function to access the network at a new location smoothly. Additionally, it ensures users with an uninterrupted and reliable connection, communication, and service [2]. In B5G systems, the importance of mobility management is significantly increased because many applications are very connectivity-sensitive to the networks [3].

HO is one of the main functions of mobility management. In the ideal case, it enables the user equipment (UE) to move and change its connection to a new serving cell seamlessly without interruption. In B5G networks, HO is becoming more challenging, especially with high-frequency bands (such as mmWave) and ultra-dense small cells. The use of mmWave leads to reduced cell coverage and a larger number of deployed small BSs, thus further increasing the number of HO compared to legacy networks. This affects the network performance regarding signaling load, connectivity and throughput. Therefore, the HO process in a new mobile network must be fast and the connection should be more seamless as many applications require very low latency and zero link failures. Thus, developing a handover technique that can provide a seamless and robust HO process is a critical challenge that needs to be solved in B5G networks. Different solutions have been proposed and developed to solve mobility issues in 5G and B5G networks [4,5,6,7].

Mobility Robustness Optimization (MRO) is one of the significant functions that have been introduced in 4G and 5G mobile networks to optimize Handover Control Parameters (HCPs). It is one of the prominent techniques that focuses on HCPs, such as time-to-trigger (TTT) and HO margin (HOM). Adjusting HCPs properly is the main factor in providing an optimum HO process.

Several existing techniques have been developed to improve the optimization process and estimate more optimum HCPs. The work presented in [8] aims to decrease the effect of HO ping-pong (HOPP). The results illustrate that the HOPP is greatly decreased compared to the literature techniques. However, the algorithm did not efficiently lower the HO probability (HOP); it is still high. Studies in [9] and [10] focused on HO management and enhancing HO delay but did not consider HOPP and HOP. Furthermore, schemes in [11] and [12] succeed to improve particular HO performance metrics, but it is at the cost of deterioration of other HO metrics. More related works are discussed in the next section. Consequently, HO management improvements are still required to enhance the overall system performance.

This paper extends our previous work [13], which focused on HOP and HOPP effect reductions. In general, the main contributions of this paper are summarized in the following points:Proposition of a Robust Handover Optimization Technique with Fuzzy Logic Controller (RHOT-FLC) to automatically adjust the HCPs more efficiently for 5G and B5G mobile networks. In other words, we developed a fuzzy-based algorithm that utilizes the advantages of the FL system to automatically adjust the TTT and HOM simultaneously. The proposed technique exploits the UE’s information such as RSRP, RSRQ, and speed to adapt the TTT and HOM as the system outputs.Evaluation of system performance in terms of HO probability (HOP), HO failure (HOF), HO ping-pong (HOPP) effect, HO latency (HOL), and HO interruption time (HIT), with different mobility speed scenarios. The RHOT-FLC technique aims to improve the HO performance in a B5G mobile system in terms of these mentioned KPIs.Comparison and performance analysis of the RHOT-FLC technique with various techniques from the literature, such as the conventional HO method (Conv), Fuzzy Logic Controller (FLC) [14] algorithm, and another competitive algorithm by Silva et al. (Slv) [15].

The rest of the paper is structured as follows: Section 2 presents recent related works, while Section 3 introduces the proposed technique (RHOT-FLC) and system model. Section 4 describes the simulation environment and performance metrics, then Section 5 presents the simulation report and performance analyses. Finally, Section 6 concludes our paper.

## 2. Related Work

In 2019, Souza et al. [8] proposed a heuristic for HO based on the Analytic Hierarchy Process (AHP) and Technique for Order of Preference by Similarity to Ideal Solution (TOPSIS), named H2ATF. A ranking of eNB according to priority was generated by adopting these elements: (1) AHP for defining criteria weights, (2) TOPSIS for ranking the chosen destination cells, and (3) an adaptive hysteresis fuzzy inference system to perform the calculation using parameters with direct impact on HO. MATLAB simulations were implemented with various small-cell eNodeBs (eNBs) and fixed macrocells to validate the efficacy of the proposed heuristic scheme. According to the findings, determining the optimal HO time was possible with a suitable antenna. Furthermore, a reduction of up to 43% in HO ping-pong, a commonly used metric to assess HO heuristics, was also achieved. The study emphasized that new parameters for HO decisions, such as the direction, flow type, and antenna load, should be further explored. Aside from that, alternative mobile network design and novel computational intelligence approaches (e.g., unique fuzzy approach, genetic algorithm, and clustering) may be examined to optimize the decision-making process. However, since it ignores the TTT, the results affect the radio link failure (RLF), which affects the HO performance.

In 2013, Muñoz et al. [14] investigated HOM and TTT to evaluate the HO performance for different UE speeds and system load systems. Moreover, the authors proposed an FLC optimization technique to adjust the HOM. The technique was set up with three different configurations of FLC and evaluated to provide the optimal key performance indicators (KPIs): call-drop ratio and HOP. Although the technique achieved a good HO performance in the call-drop ratio and HOP, it does not provide exemplary performance in the other HO’s KPIs, such as RLF and HOPP.

In 2018, Silva et al. [15] introduced a fuzzy logic-based scheme that leverages user velocity and radio channel quality to self-optimize a hysteresis margin for HO decision to address mobility management issues. The work targets future networks with a large number of small cells while enabling smooth connectivity. MATLAB was utilized to assess the proposed solution while considering the situation of a 1000 m × 1000 m area, where 2 eNBs, up to 200 small cells eNBs, and 50 UEs were placed in this region. Based on the findings, the study concluded that, in all cases, the proposed algorithm efficiently restricted and kept its ping-pong effects below 1%. Moreover, the HOF ratio and the HO occurrences were significantly decreased when compared to previous methods, particularly for cases with many small cells. The dwelling time of users in the small cells was also comparable to similar schemes. As for future work, the study emphasized focusing on signal-level prediction for the throughput gain ascertainment and investigating potential enhancements for HO performance metrics. However, the algorithm is limited to a certain mobility speed of up to 80 km/h, which is considered moderate.

In 2017, Nguyen et al. [16] proposed a distributed MRO algorithm to reduce HOFs caused by RLFs by modifying trigger time and offset parameters. The algorithm categorized three HOF types (wrong cell, too late, and too early) and optimized three HO parameters simultaneously based on the prevailing failure. Furthermore, the algorithm considered HOFs in each nearby cell and modified HO parameters separately. Simulations were performed using the Long-Term Evolution (LTE) module in the ns-3 network simulator to validate the proposed MRO algorithm [17]. A one-tier small-cell network with seven small eNBs was also studied, with a 30 m inter-site distance. According to the simulation findings, the algorithm effectively discovered the best HO parameters and surpassed prior techniques. Furthermore, the proposed method produced the fewest ping-pongs of the algorithms studied. However, it only focused on RLF and HOPP and ignored discussing other important HO KPIs, such as HOP probability, HIT, throughput, and cell edge spectral efficiency.

In 2018, Hegazy et al. [18] introduced a method to optimize two opposing HO problems (i.e., RLFs and ping-pongs) utilizing fuzzy Q-learning. The first must reduce the HO margin to reduce the too-late HO, whereas the second must raise the HO margin to reduce the needless signaling. Users in the network were classified into four groups based on their speed and data traffic usage. As a result, keeping general HO issues within acceptable levels can enhance the user experience. Fuzzy Q-learning was used with distinct initial candidate fuzzy actions for each user group. LTE-Sim [19] was used to assess the proposed algorithm’s efficacy. However, these two existing research methods do not apply any optimization method. The findings indicated that the suggested technique offered the optimum performance for each user group corresponding to the most chosen metric, either decreasing ping-pongs or lowering RLF. Furthermore, whether the number of users was cut by half or doubled, the proposed algorithm was proven to be resistant against changes in the number of users. Furthermore, not all KPIs have been investigated.

In 2019, Liu et al. [20] proposed a fuzzy-TOPSIS-based HO algorithm to reduce the ping-pong effect and number of HOs. The proposed algorithm leveraged the benefits of fuzzy logic and TOPSIS and utilized the received signal-strength intensity (RSSI) and signal-to-noise ratio (SNR) as the HO criteria. According to the MATLAB simulation results, the proposed HO algorithm could effectively decrease the ping-pong rate and the number of Hos when compared to conventional RSSI-based HO approach and classical multi-attribute decision-making (MADM) HO methods, such as simple additive weighting (SAW) and basic TOPSIS. Nevertheless, HO performance such as HOF, and HOL was not evaluated to show the robustness of the algorithm with various KPIs. 

In 2019, Goyal et al. [21], the authors presented a scheme for choosing the optimal network in a heterogeneous network situation to preserve overall network quality of service (QoS) by employing multiple criteria: velocity, received signal strength (RSS), dwell time, bandwidth, cell radius, load on eNB, and power transmission. The best-ranked network with the lowest HOF was chosen using the fuzzy AHP method. The findings were statistically evaluated and revealed that the optimum network with a lower HOF rate may enable uninterrupted communication. Nonetheless, the study did not consider other aspects that may impact UE and the network during HO, such as UE energy consumption and packet loss.

In 2019, Alhammadi et al. [22] proposed a weighted fuzzy self-optimization (WFSO) approach for optimizing HCPs. The HO choice in this method was based on three key attributes: SINR, the traffic load of serving and target BS, and UE velocity. To increase HO performance, the self-optimized HCPs (i.e., HOM and TTT) were modified based on the present status of these attributes. MATLAB was used to comply with the third-generation partnership project (3GPP) HetNet mobility simulation methodology [23]. Based on the results, HOPP, RLF, and HOF rates were considerably reduced using the developed WFSO compared to other benchmark methods. However, the algorithm did not provide optimum performance in all HO KPIs.

Finally, and most recently, Lema et al. [24] assessed HO performance in 2021 by optimizing the HCPs, such as TTT, HO offset, and HO margin. A well-known heuristic algorithm known as particle swarm optimization (PSO) was also employed. In addition, the performance of the networks could be improved by a self-organized network (SON) through the automation of mobility load balancing (MLB) and PSO. The results were validated using MATLAB Software. Compared to standard HO performance enhancement approaches, the simulation results revealed that the PSO and MLB-based HOF and HOPP were considerably improved. However, the probability of the HOF is highly increased as the UE’s speed increases. Additionally, not all KPIs were considered in this study.

Although the aforementioned works have attained HO performance enhancement in terms of different HO KPIs, an optimum solution that can provide optimum HO performance for all HO KPIs is still required, and the developers and researchers are working on delivering robust HO techniques that overcome the HO issues. Table 1 summarizes the related works.

## 3. Proposed System

This section explains the HCP optimization and gives a general overview of the FLC system. It also presents the proposed RHOT-FLC technique, including its system process.

### 3.1. Overview

Two important HCPs are TTT and HOM. Both of these have a significant effect on HO. HOM is a variable that indicates the threshold of the difference between the strength of the signal received at the serving base station and that of signals received at target BSs. TTT indicates the essentially vital time interval for meeting HOM conditions. Two criteria must be met for the execution of HO: The first criterion requires the RSS of the serving BS to be outshined by the RSS of a potential cell. In contrast, the second criterion involves the fulfillment of the first condition within the time specified in the TTT parameter [25].

Ensuring reliable and stable communication throughout the UE mobility is one of the critical challenges facing the practical implementation of the B5G networks. One of the solutions to design and improve the HO robustness is using the HCP-based techniques, which is considered a self-optimization technique. HCPs (TTT and HOM) are vital criteria in HO decision techniques. Therefore, they must be adapted and optimized very well to provide optimum HO performance. For example, increasing the value of TTT in a high-speed scenario increases the RLF probability, which degrades the overall system performance. On the contrary, adjusting TTT with a small value in a low mobile speed scenario increases the HOPP probability. Similar to the HOM, adjusting TTT with different values affects the HO performance accordingly.

A fuzzy logic system is a mathematical system based on fuzzy logic that analyzes analog input values in continuous logical variables values between 0 and 1. A fuzzy logic controller (FLC) generally consists of three main stages: fuzzification, inference, and defuzzification. During fuzzification, numerical input variables are converted into membership functions. The systems’ output has linguistic relations with the system’s inputs. These relations are called rules, and the output of each rule is referred to as a fuzzy set. More than one rule is used to increase conversion efficiency. The inference is the process whereby the fuzzy output sets of each rule are combined to make one fuzzy output set. Afterward, the fuzzy set is defuzzified to a crisp output in the defuzzification process.

FLC has been widely applied to mobile network parameter optimization as in studies by [14,15,22,26,27,28,29,30,31] for HO. FLC has the advantages of working with imprecise inputs, not needing an accurate mathematical model, and handling nonlinearity, as proved by the results of this research. In mobility management, FLC is used to adjust and adapt the HCP (TTT and HOM) values to provide an automatic HO process. The existing works, for example, in [15,18,20,21], applied FLC in their algorithms for HO in mobile systems and show that FLCs are very useful for automatic HO parameter optimization.

A membership function can be described, for example, by a triangular fuzzy number $\left(x\right)$ (Figure 1) consisting of triplet *a*, *b*, *c*. The fuzzy triangle function is suitable for real-time operation due to its modelling simplicity. The mathematical expression of the membership function, in this case, is as the following Equation (1):(1)$$f\left(x\right)=\left\{\begin{array}{c} 0,\;x\leq a\\ \frac{x-a\;}{b-a},\;a\leq x\leq b\\ \frac{c-x}{c-b},\;b\leq x\leq c\\ 0,\;c\leq x \end{array}\right\}\;$$

### 3.2. Robust Handover Optimization Technique with Fuzzy Logic Controller (RHOT-FLC)

We propose a robust handover optimization technique (RHOT-FLC) exploiting the FL system to enhance the HO performance in B5G networks by mitigating the HOPP rate, HOP, HOF, HIT, and HOL. The proposed technique is designed to automate the HO decision and adjust TTT and HOM. The system consists of 48 rules based on three input parameters: UE velocity, RSRP, and RSRQ. These rules are used to dynamically estimate two different outputs, TTT and HOM, in every single process. The 48 rules are formulated according to Table 2 and the triangle function, Equation (1). The mobile speeds and RSRQ are classified into four levels, while the RSRP is classified into three categories. Both TTT and HOM are divided into four categories that can enhance HO performance. Increasing the number of levels may enhance the accuracy of selecting the TTT and HOM values. However, it increases the processing time and vice versa. Therefore, the four levels are selected to compromise the system performance, accuracy and processing time. In other words, the four levels are maintained to achieve the desired HO performance, as presented in the Results and Discussions Section. The rules (cases) are formulated by adjusting the values of the TTT and HOM and evaluating the output using the considered HO performance metrics. In addition, the rules are determined based on the evaluation conducted on each rule and overall system performance results.

Many of the existing works, such as in [15,18,20,21], have demonstrated that the FL system could provide optimum HO performance. Besides that, the FL system can simultaneously optimize the two important HO parameters, which are TTT and HOM, with accurate results, as demonstrated in our results. However, designing the HO algorithm based on the FL system depends on the classifications of each input, formulating rules, and considering input parameters. The RHOT-FLC technique considers the UE’s velocity, RSRP, and RSRQ as the system input and designed 48 rules and cases. The values of the RSRP and RSRQ are adjusted according to 3GPP, Release 16 definition [32]. Although HO algorithms based on the FL system can achieve excellent HO performance, the HO time process is increased as the input parameters and rules are increased.

Moreover, the proposed system has been designed to automatically adjust both TTT and HOM concurrently, unlike many works, which focus on adjusting only one HCP, either TTT or HOM [8,15,16]. Figure 2 illustrates the proposed RHOT-FLC system, consisting of five stages. The first stage is the input stage, which consists of three measured values as updated input from Algorithm 1 (UE’s velocity, RSRP, and RSRQ). In the fuzzification stage, the crisp input values are converted to fuzzy values in the next stage. The third stage is the inference engine, in which the proposed if-then rules are applied. Next, the generated fuzzy values are converted into crisp values in the defuzzification stage. The last stage is the output stage, which is TTT and HOM. After updating the HCPs with adjusted values, the process is completed according to the Algorithm 1 process. Algorithm 1 presents the RHOT-FLC algorithm. The general steps of the proposed technique are as follows:The RSRP for all gNBs is sorted and compared to the gNBs target station. If the Equation (2) condition is not fulfilled, the HO decision is not performed. Else,
(2)$$RSRP_{target}>RSRP_{serving\;}+HOM\;$$

2.Update the system inputs with RSRP, RSRQ, and UE speed.3.Convert the inputs values into fuzzy sets and calculate the degree of each membership function, according to Equation (1) (Figure 2).4.Apply the proposed rules (48 if/then rules) for each membership.5.The TTT and HOM are updated as system output according to the three input parameters conditions. The input parameters are defined in 48 cases.6.Update the system with adapted HCPs to perform the HO decision.

**Algorithm 1:** **RHOT-FLC**

*Initialize System Parameters (B5G network)*
***if***    *t = 1 (t: simulation time)*
*Handover Decision ⟶ False*

**
*else*
**

*Measure: RSRP, RSRQ, and UE Velocity*

*Sort RSRP_all_*

***if**    RSRP _serving_ > RSRP _target_*

*Handover Decision ⟶ False*

**
*elseif*
**

*Update: RSRP, RSRQ, and UE Velocity*

*Input: Define input parameters*

*Convert input parameters to fuzzy sets*

*Calculate the degree of each rule*

*Output: Adapt TTT and HOM*

*Update the HCPs*

**
*else*
**
*RSRP _target_ > RSRP _serving_ + HOM*

***if** trigger time = TTT*

*Handover Decision ⟶ True*

**
*else*
**

*Handover Decision ⟶ False*

**
*end*
**

**
*end*
**

**
*end*
**



**Figure 2 sensors-22-06199-f002:**
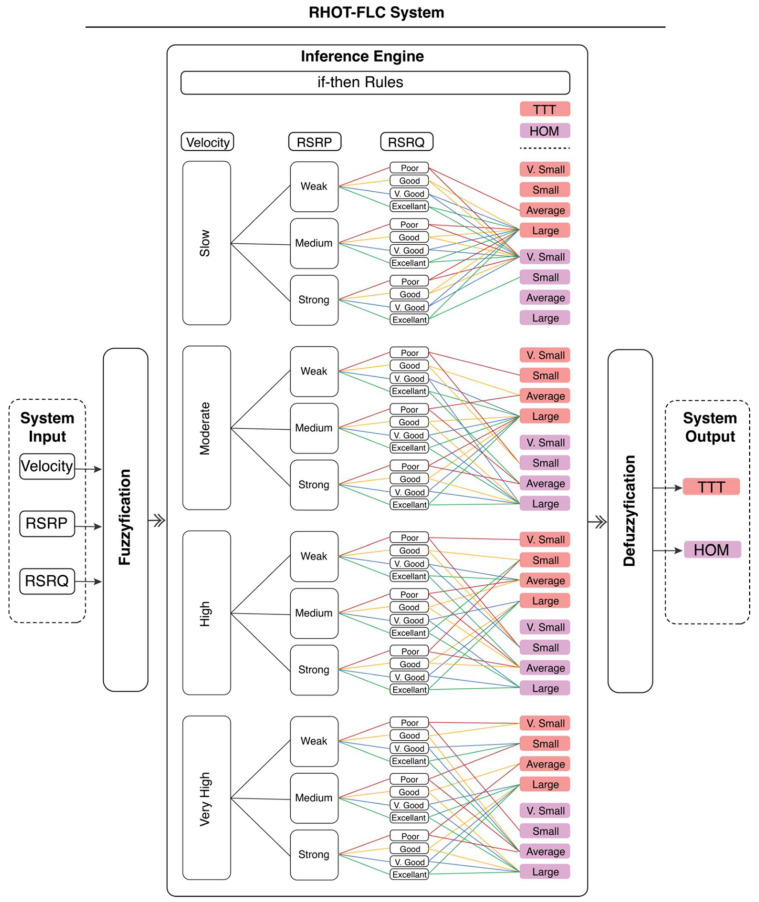
RHOT-FLC proposed system.

## 4. System and Simulation Model and Performance Metrics

This section presents the simulation model, including the network deployment scenario and parameters. It also provides the HO performance metrics used to evaluate the system.

### 4.1. Network Deployment Scenario

The simulation environment has been developed in MATLAB 2020b to simulate the B5G network considering microcells and urban area scenarios. The network layout consists of sixty-one gNBs with three sectors for each cell that are deployed in a $\left(3000\times 3000\right){\;m}^{2}$ simulation environment, and the distance between two BSs is 400 m (each BS covers 200 m).

The UEs move in a straight way within eight directions [*N, NE, E, SE, S, SW, W,* and *NW*] within the simulation environment and pass-through BSs with five different scenario speeds, which are 20 km/h, 40 km/h, 80 km/h, 120 km/h, and 160 km/h. Each UE was randomly initiated to move within the possible directions straightly with different mobile speeds, until it reached the edge of the defined area and changes its movement direction randomly within possible directions.

The considered assumption for the use-case scenario is ultra-high-definition video (UHDV) streaming, such as 4K video streaming. Video streaming is one of the prominent applications that all users use, and its demands are dramatically increased. According to Ericsson forecasts, video streaming will account for 76% of mobile traffic by 2025 [33]. Moreover, by the year 2023, it is expected that around 75% of mobile data traffic will be produced by video applications [34]. As a result, ensuring the quality of video streaming via B5G networks may significantly influence passengers’ travel experiences.

In this study, 200 UEs’ traffic per simulated cell has been proposed to be generated and distributed randomly throughout the coverage area. Then, it is changed dynamically and randomly in each simulation cycle. This assumption is considered to simulate a real network scenario as high traffic negatively affects the overall system performance. Once the cell traffic is increased, the HOP is increased to balance cell loads. This may lead to high degradation in the network performance in terms of HOPP, RLF, interruption time, throughput and spectral efficiency. Ten UEs were chosen to be measured in this study to investigate the HO performance for different KPIs compared with the competitive methods (such as the driving test in real life, where only one or two users are used to evaluate the network). Thus, when the ten UEs move within the cells, their performance will be impacted negatively or positively based on the cell loads of each serving cell. Figure 3 illustrates the deployment scenario.

### 4.2. Simulation Parameters

The channel model and simulation parameters are adjusted according to 3GPP Release 16 [35,36,37,38], as 3GPP has defined mmWave frequency with 28 GHz as a prominent candidate frequency band considered in a B5G system to meet the increasing demand for user data throughput. The main simulation parameters are presented in Table 3.

### 4.3. Performance Metrics

#### 4.3.1. Handover Probability (HOP)

HOP is the probability of HO when the UE moves from one cell to another. Likewise, HOP represents the percentage of HOs occurring. One of the cases that increases the HOP is the HOPP.

Increasing HOP leads to an increase in the system complexity and affects the overall performance. The average HOP in the network in each simulation time and overall UE is calculated as the following expression:(3)$$HOP=\frac{\sum_{i=1}^{N_{UE}}\;P_{i}\left(HO\right)}{N_{UE\;}},\;$$
where $N_{UE}$ is the number of UEs. Each UE is moving in a random direction, which means that each UE has a different location, different signal strength, and different BS coverage. Therefore, each UE may have a different HOP.

#### 4.3.2. Handover Failure (HOF)

HOF refers to unsuccessful HO from serving gNB to a target gNB. The inappropriate setting of HCPs increases the HOF rate, thus leading to a rise in the RLF probability. In other words, HCPs play the main role in increasing or decreasing the HOF rate. Further, HOF happens due to four scenario cases: too-early HO, too-late HO, wrong HO, and ping-pong HO. The too-early and too-late HO occurs due to incorrect TTT. A low value of TTT causes too-early HO, whereas a high value of TTT causes too-late HO. In the case of the wrong HO, the HOF occurs when the UE is handed over to the wrong cell. Ping-pong HO happens when the UE is at the border of two or more neighbor cells, and UE changes its connection gNB to another gNB in a concise time. The HOF probability is calculated as follows:(4)$$HOFP=\frac{N_{HOF\;}}{N_{HO\;}},\;$$
where $N_{HOF}$ is the total number of HOF, and $N_{HO}$ number of (failure HO + successful HO).

#### 4.3.3. Handover Ping-Pong (HOPP)

The HOPP is the frequent HO that happens between two neighbor cells. The frequent movement of the UE between the boundaries of the two neighboring cells results in a ping-pong effect due to high signal fluctuations.
(5)$$\bar{HOPP}\left(\%\right)=\frac{N_{HOPP}}{N_{HO}},\;$$
where $\bar{HOPP}$ is the HOPP probability, $N_{HOPP}$ represents the number HOPP, and $N_{HO}\;$ is the number of (failure HO + successful HO).

The instantaneous average HOPP probability $\left(\bar{HOPP}\right)$ over all UEs can be given as follows:(6)$$\bar{HOPP}=\frac{\sum_{i=1}^{N_{UE}}N_{HOPP}\left(i\right)}{N_{UE}},$$
where i is the corresponding index of the measured user, and $N_{UE}$ is the total number of measured UEs. Moreover, each UE moves in a random direction, which means that each UE has a different location, different signal strength, and different BS coverage. Therefore, each UE may have a different HOPP.

#### 4.3.4. Handover Latency (HOL)

HOL is an essential measurement of system performance. According to 3GPP [39], HOL is when UE has received the HO command from the serving gNB to complete the HO process to the target gNB. In other words, the HOL is the total time that is taken for the HO execution stage. B5G networks require very low latency, up to 1 ms, as several applications are sensitive to the time; for example, autonomous vehicles require ultra-low latency to avoid accidents.

#### 4.3.5. Handover Interruption Time (HIT)

The HO Interruption Time, or HIT, is the instant during the execution of HO when there is an interruption in the user data exchange between the source and target cell by the mobile terminal. This suggests that HIT is the minimum time supported by a cellular network during HO. The mobility interruption time ranges from 30 to 60 ms for a 4G LTE deployment [40]. The factors affecting this interruption time include HO conditions and radio conditions. The 3GPP community aims to reduce the interruption time to allow the effective use of 5G wireless technologies in future applications. The interruption time can be precisely reduced to around zero ms in the B5G networks [41].

## 5. Results and Discussions

This section provides the simulation study results for RHOT-FLC and the three competitive algorithms from the literature: Conv method, FLC, and Slv. The proposed technique is evaluated using five important HO performance metrics (HOP, HOF, HOPP, HOL, and HIT) and assessed in five mobile speed scenarios (20, 40, 80, 120, and 160) km/h, as explained in the previous section. Furthermore, the algorithms are validated by using simulation with B5G networks. The presented results illustrate the average measured values of the 10 UEs in each mobile speed scenario and overall simulation times. The performance of each UE is evaluated and collected in every simulation cycle (50 ms) for each mobile speed. Each HO performance is evaluated individually.

The proposed technique is compared with three competitive algorithms, which are the techniques presented [15] (denoted as Slv in the results and figures), a conventional HO algorithm based on the quality of signal criterion plus HOM (denoted as Conv in the results and figures), and FLC [14] (denoted as FLC in the results and figures). The competitive algorithms are chosen because they focus on mobility management and MRO while having similar techniques as in FLC and Slv. Further details are presented in Table 1. Meanwhile, these three techniques have been explained and investigated in more detail compared to the other techniques in the literature. To ensure fairness in the comparison, we used the same simulation parameters, scenario, and environment for the proposed and competitive algorithms.

### 5.1. Handover Probability (HOP)

Figure 4 shows the average HOP overall mobile speeds and simulation times for 100 s. The result indicates that RHOT-FLC achieved the lower HOP for all mobile speeds and simulation times with less than 3.6%. The consistency of RHOT-FLC provides a substantial reduction in HOP, indicating the appropriate adjusting of HCPs of RHOT-FLC. The other algorithms provided higher HOP and higher fluctuations, which means insufficient accuracy in adjusting the HCPs of the algorithms.

Figure 5 presents the average probability of HO for all algorithms and overall mobile speeds. The figure shows that RHOT-FLC significantly reduces the HOP compared to the other algorithms, Conv, FLC, and Slv. RHOT-FLC has obtained the lowest HOPs of less than 3.6%, while the Conv, FLC, and Slv achieved HOPs of 37%, 25.7%, and 74%, respectively.

The number of HOs is increased drastically in B5G networks due to the requirements of B5G, which aims to support a massive number of devices per area and uses the mmWave operating band, which has a very small coverage area of up to 200m. Therefore, many small cells are required to be deployed in a small area, which increases the HOP. However, RHOT-FLC shows a significant reduction in HOP up to 90%, 86%, and 95% compared to the Conv, FLC, and Slv, respectively. The reduction in HOP decreases HOF probability, enhancing HO performance.

### 5.2. Handover Failure (HOF)

Figure 6 illustrates the results of HOF probability for all algorithms and overall mobile speed scenarios. The results depict that the lowest HOF probability is attained by RHOT-FLC by less than 0.19%, while the HOFs of the Conv, FLC, and Slv are 2%, 1.4%, and 4.6%, respectively. RHOT-FLC has considerably reduced HOF by 90.5%, 86.4%, and 95.8% compared to Conv, FLC, and Slv, respectively. HOF was caused due to the failure of UE to connect to the target gNB. However, RHOT-FLC can reduce the HOF probability and help to improve the overall HO performance even in high-mobile-speed scenarios. Moreover, the results in all algorithms suggest the behavior of HOF probability reflects the HOP.

### 5.3. Handover Ping Pong (HOPP)

Figure 7 depicts the HOPP probability with different mobile speeds for selected simulation times (50 s). RHOT-FLC achieved lower HOPP probability than the other algorithms at all mobile speeds. The results give an additional view that RHOT-FLC reacts better with the speeds than the other algorithms and is preserved at a low rate. This justifies the robustness of RHOT-FLC. Furthermore, the competitive algorithms obtained higher HOPP and fluctuated probability. For instance, the Slv algorithm obtained the highest HOPP and probability fluctuations along the selected time. This phenomenon may be justified due to the inappropriate setting of HCPs. Furthermore, adjusting the HCPs with high-level values caused a high probability of the RLF.

HOPP is an essential HO performance metric that represents the unnecessary HO. Figure 8 illustrates the probability of HOPP with different speed scenarios. Overall, the HOPP ratio for all algorithms gradually decreases as the speed increases. This condition due to UE is moving straight, which means that the UE at low speeds stays at the cell’s edge longer than at higher speeds. Additionally, assigning HOM and TTT with low values results in an increase in the HOPP. Therefore, auto HO techniques aim to adjust the HOM and TTT perfectly to preserve optimal HO performance. However, it is seen that RHOT-FLC is superior to all algorithms used for comparison. RHOT-FLC achieved the lowest HOPP probability of less than 1.9%, as compared to 30%, 20%, and 64% in Conv, FLC, and Slv, respectively. The best performance is achieved at 160km/h for all algorithms by 26%, 16%, 55%, and 0.45% for Conv, FLC, Slv, and RHOT-FLC.

Figure 9 shows the average HOPP probability for all algorithms and overall mobile speeds and simulation times. It can be seen that RHOT-FLC attained the overall average HOPP performance for all mobile speeds with an average HOPP probability of 1.9% and 30%, 20, and 64% in Conv, FLC, and Slv, respectively. Therefore, RHOT-FLC substantially improves HOPP up to 93.6%, 90.5%, and 97% compared to Conv, FLC, and Slv. The optimum HO performance shown in HOPP by RHOT-FLC explains that RHOT-FLC can perfectly adjust the HCPs. The reduction in HOPP decreases the HOP, thus decreasing the HOF probability. Therefore, this improves the overall HO performance.

### 5.4. Handover Latency (HOL)

The average HOL for all algorithms and overall mobile speeds is illustrated in Figure 10. The figure shows that the RHOT-FLC significantly decreased the average HOL up to 3.7 ms. At the same time, the other algorithm, Conv, FLC, and Slv attained average HOL up to 35.9 ms, 25 ms, and 70.8 ms, respectively. Thus, RHOT-FLC provides the best HOL performance with an average 89.7%, 85%, and 94.7% improvement in HOL compared to Conv, FLC, and Slv, respectively. Thus, RHOT-FLC can improve the overall HO process, which enhances the overall HO performance.

### 5.5. Handover Interruption Time (HIT)

The result of HIT is presented in Figure 11. The figure provides the average HIT for all algorithms and overall mobile speeds. The figure shows that RHOT-FLC obtains the lowest HIT by 1.8 ms compared to 18.5 ms, 12.8 ms, and 37 ms by Conv, FLC, and Slv, respectively. In other words, it is seen that RHOT-FLC obtained the best HIT performance with a HIT reduction of 90.5%, 86%, and 95% compared to Conv, FLC, and Slv, respectively. Moreover, the HIT performance achieved by RHOT-FLC is substantially enhanced, which can be considered an optimum performance of the B5G networks.

In summary, as the results demonstrated, RHOT-FLC outperforms all the competitive algorithms in all considered HO performance metrics, HOP, HOF, HOPP, HOL, and HIT. This explains that the proposed technique can adjust the HCPs (TTT and HOM) efficiently and appropriately. Furthermore, simultaneously adjusting both TTT and HOM properly leads to substantially reducing the HOP, HOF, HOPP, HOL, and HIT, thus significantly enhancing the HO performance.

Regarding the complexity of the competitive algorithms and RHOT-FLC, they have a slight difference. RHOT-FLC, FLC, and Slv algorithms are FL-based techniques, but the RHOT-FLC technique may have higher computational complexity as a result of the technique being designed to adjust both HOM and TTT at the same time, and its rules are formulated to support low and high speeds, which leads to increasing the algorithm’s time execution. The FLC and Slv algorithms was designed to adjust the HOM only, while the conventional method (Conv) has the lowest complexity because the algorithm basically fixes the values of the TTT and HOM at certain values based on the RSRP only; there is no optimization or adapting process. Overall, the complexity of the competitive algorithms and RHOT-FLC can be sorted from the lowest to the highest time complexity as follows: Conv, FLC, Slv, and RHOT-FLC. Nevertheless, the RHOT-FLC significantly enhanced the HO performance as compared with the competitive algorithms.

Table 4 summarizes the average performance of HO for all algorithms and the overall improvement of RHOT-FLC compared to the state of the art. Moreover, from the following tables, it can be noticed that RHOT-FLC dramatically enhances the overall HO performance by lowering the HOP, HOF, HOPP effect, HOL, and HIT. Moreover, RHOT-FLC has achieved an overall improvement of 90%, 86%, and 95% compared to Conv, FLC, and Slv, respectively. The excellent HO performance obtained by RHOT-FLC indicates that RHOT-FLC can appropriately adjust the HCPs.

## 6. Conclusions

This paper proposed RHOT-FLC to optimize the HCP (TTT and HOM) parameters dynamically. The FLC-based technique exploits UE information, such as RSRP, RSRQ, and UE’s speed as the system inputs. Different HO KPIs, such as HOP, HOF, HOPP, HOL, and HIT, are considered to verify the technique. Furthermore, the technique is investigated considering different mobile speed scenarios, which are 20, 40, 80, 120, and 160 km/h. As the results show, the RHOT-FLC technique significantly improves the HO performance by considerably reducing the probabilities of HOP, HOF, HOPP, HOL, and IT. The overall improvement of RHOT-FLC is up to 90.76%, 86.78%, and 90.5%, compared to the literature Conv, FLC, and Slv algorithms, respectively. The optimum results provided by RHOT-FLC indicate that the HCPs are adequately adjusted. Therefore, it enhances the overall HO performance.

Additional HO performance metrics, such as RLF and frequency efficiency with higher mobile speed scenarios, will be tested in future works in addition to different deployment networks consideration.

## Figures and Tables

**Figure 1 sensors-22-06199-f001:**
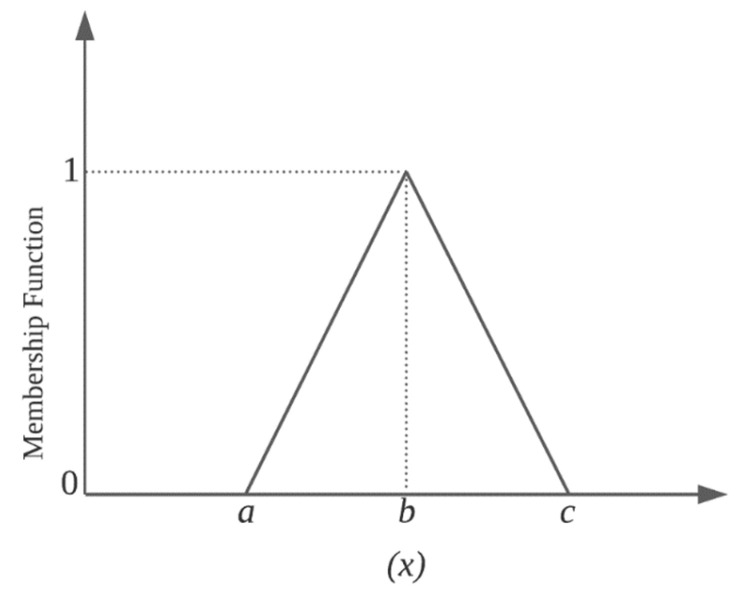
A common membership function of an FL system.

**Figure 3 sensors-22-06199-f003:**
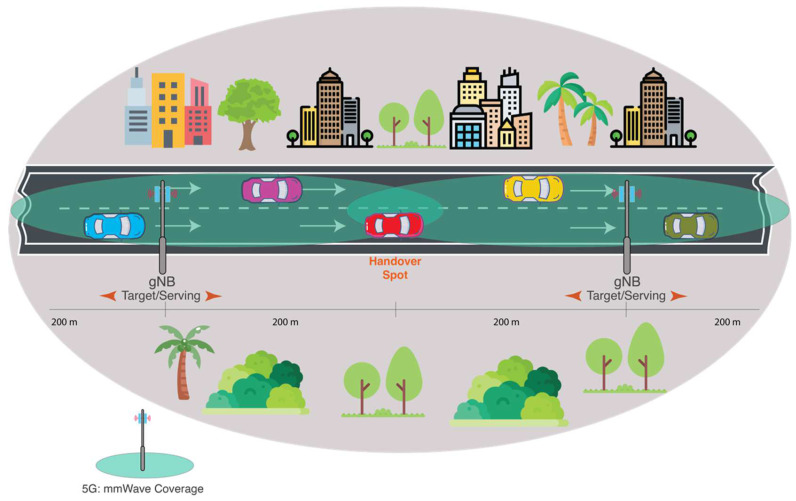
B5G network deployment scenario.

**Figure 4 sensors-22-06199-f004:**
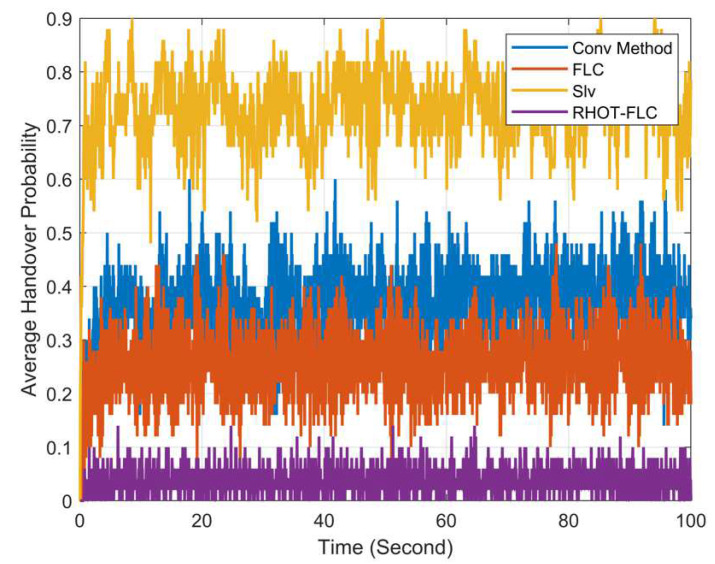
Average HO probability for overall mobile speeds and simulation times.

**Figure 5 sensors-22-06199-f005:**
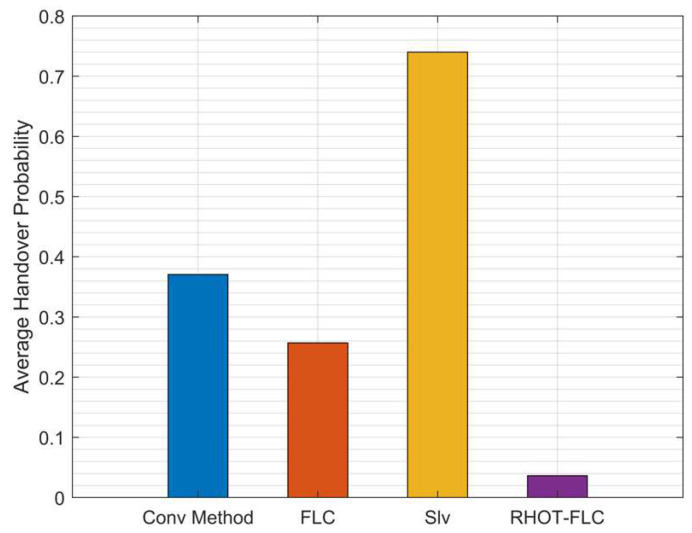
Average HO probability for overall mobile speed scenarios.

**Figure 6 sensors-22-06199-f006:**
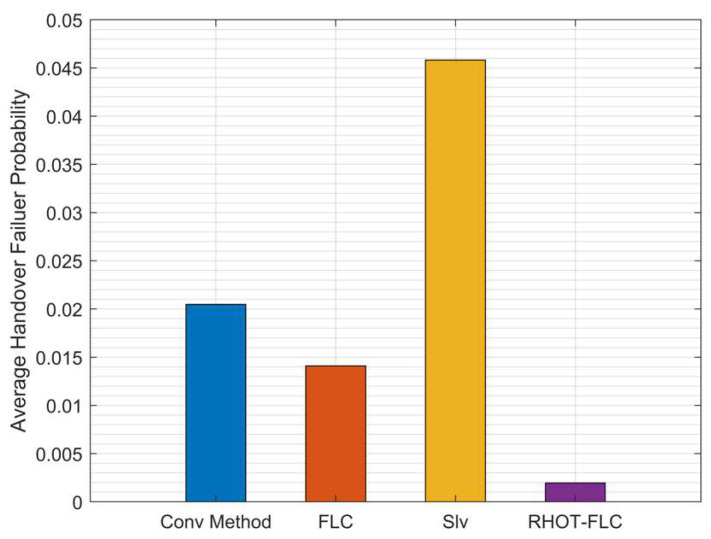
Average HOF probability for overall mobile speed scenarios.

**Figure 7 sensors-22-06199-f007:**
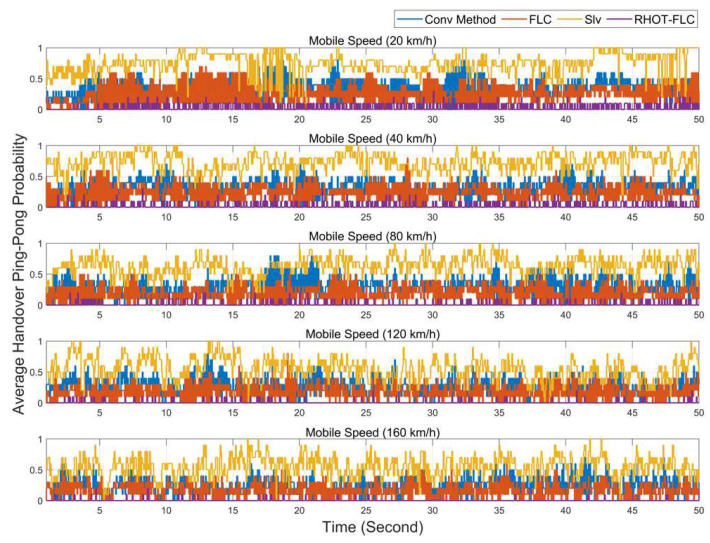
Average HOPP probability for different mobile speeds.

**Figure 8 sensors-22-06199-f008:**
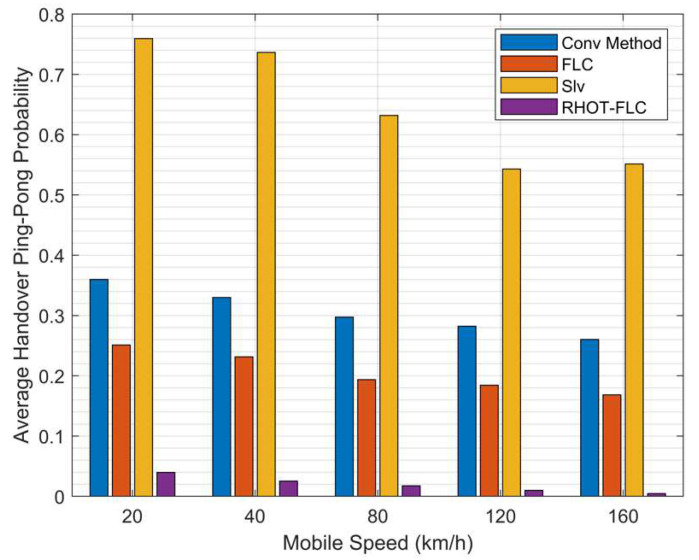
HOPP probability with different mobile speed scenarios.

**Figure 9 sensors-22-06199-f009:**
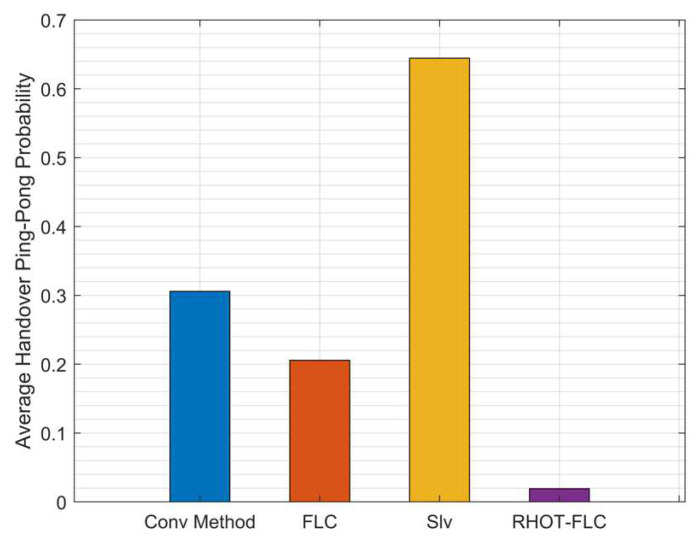
Average HOPP probability overall mobile speeds scenarios.

**Figure 10 sensors-22-06199-f010:**
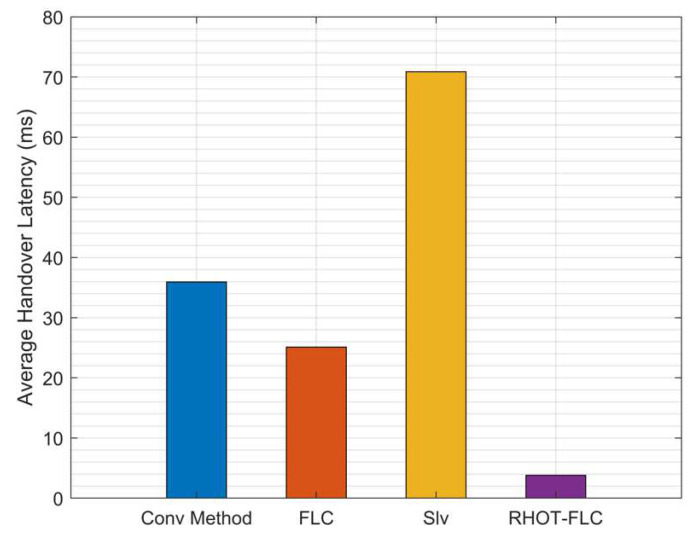
Average HOL overall mobile speeds scenarios.

**Figure 11 sensors-22-06199-f011:**
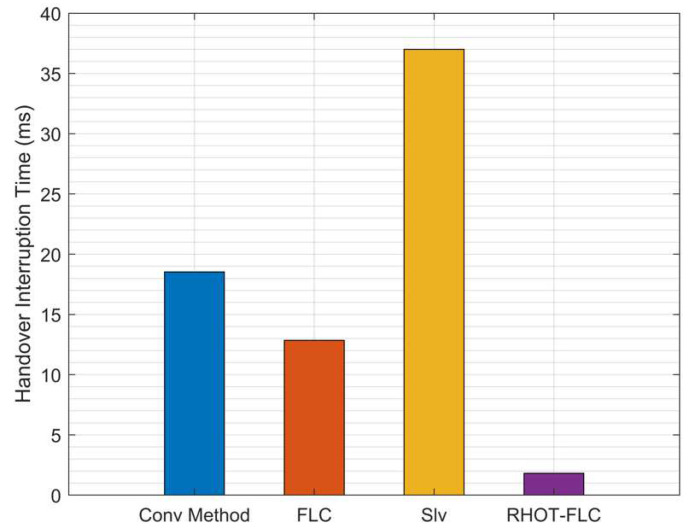
Average HIT overall mobile speeds scenarios.

**Table 1 sensors-22-06199-t001:** Summary of Related Works.

Ref.	Problem	Solution	System	Optimization Parameters	Performance Metrics
[8]	Deterioration in the provided quality of services due to a high number of HOs because of failure to rank the priority of BS.	Self-optimization based on AHP-TOPSIS-Fuzzy to select the target cell	Small Cells/HetNets	HOM	HOPP and HOF
[14]	Providing automated operation for Self-Organizing Networks (SONs).	FLC-based technique that adaptively adjusts HOM	LTE	HOM	Call drop ratio and HO ratio
[15]	A high number of HOs in networks of a large number of small cells	Self-optimization algorithm based on FL exploiting users’ speed and radio channel quality to adjust HOM	Dense Small-Cell Networks	HOM	Number of HO, HOF ration, and HOPP
[16]	Occurring HOF as a result of RLF, which reduces the system’s performance	MRO algorithm to adjust TTT and offset according to HOF reason	4G/Small Cells Networks	TTT	RLF and HOPP
[18]	RLF and HOPP for users using real-time traffic	MRO algorithm based on fuzzy Q-learning to adjust HOM	LTE	HOM	RLF and HOPP
[20]	Increased HOPP, number of HOs, unnecessary HOs, and frequent HOs due to deployment of a massive number of BSs	Self-optimization algorithm based on FL-TOPSIS Handover Decision-making Algorithm	4G	HOM	Number of HO and HOPP
[21]	Increasing HOF affects the QoS of the system	Fuzzy AHP-based technique that correctly selects the optimal network among the available networks as a target network with HO	LTE/ HetNets	−	HOF
[22]	Increasing the number of HOs increases the HOPP and HOF due to the deployment of a massive number of BSs	Self-optimization based on WFSO to adapt HOM and TTT	4G/5G HetNets	HOM and TTT	RLF, HOPP, and HOF
[24]	Degradation in QoS due to a high probability of HOF and HOPP	Self-optimization based on PSO to adjust HOM and TTT	4G	HOM and TTT	HOPP and HOF

**Table 2 sensors-22-06199-t002:** Membership values for input and output.

**Input**	**Degree**	**Range**
Velocity	slow	0 to 30 km/h
	moderate	25 to 70 km/h
	high	65 to 135 km/h
	very high	130 to 160 km/h
RSRP	weak	−160 to −95 dBm
	moderate	−100 to −73 dBm
	strong	−80 to −20 dBm
RSRQ	poor	−60 to −18 dB
	good	−22 to −12 dB
	very good	−14 to −6 dB
	excellent	−10 to +20 dB
**Output**	**Degree**	**Range**
TTT	very short	0 to 220 ms
	short	210 to 380 ms
	average	370 to 520 ms
	large	510 to 640 ms
HOM	very low	0 to 0.3 dB
	low	0.2 to 0.5 dB
	average	0.4 to 0.8 dB
	high	0.7 to 1 dB

**Table 3 sensors-22-06199-t003:** Simulation parameters.

Parameter	Values
Environment	Micro cells, urban area, B5G networks
Cell Layout	Hexagonal grid
Simulation Area (m)	$\left(3000\times 3000\right)$
Number of gNB	61
Number of Sectors	3
Cell Radius (m)	200
Maximum Number of UE per Cell	200/cell
Maximum Number of PRB per UE	2500
Number of Measured UE	10
Carrier Frequency $\left(f_{c}\right)$(GHz)	28
System Bandwidth (MHz)	500
White Noise Power Density (dBm/Hz)	−174
Path Loss	$PL_{3GPP,UMi}=35.3log_{10}\left(d\right)+22.4+21.3log_{10}\left(f_{c}\right)-0.3\left(h_{UE}-1.5\right)$
Shadow Fading (dB)	7.82
gNB Hight (m)	10
UE Hight $(h_{UE}$) (m)	1.5
UE Speeds (km/h)	(20, 40, 80, 120, 160)
UE Power (dBm)	23
Transmission Power (dBm)	35
Mobility Model	Straight-way within 8 possible directions [*N, NE, E, SE, S, SW, W,* and *NW*]
HO Decision	Equation (2)
TTT (ms)	Adaptive: 0−640
HOM (dB)	Adaptive: 0−1

**Table 4 sensors-22-06199-t004:** Average HO performance for all algorithms and overall improvement of RHOT-FLC as compared to the competitive algorithms.

KPI	Conv	FLC [14]	Slv [15]	RHOT-FLC
HOP (%)	37	25.7	74	**3.6**
HOF (%)	2	1.4	4.6	**0.19**
HOPP (%)	30	20	64	**1.9**
HOL (ms)	35.9	25	70.8	**3.7**
HIT (ms)	18.5	12.8	37	**1.8**
RHOT-FLC Overall improvement (%)	90.76	86.78	95.5	−

## Data Availability

Not applicable.

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
