# Peer review of "Robust Handover Optimization Technique with Fuzzy Logic Controller for Beyond 5G Mobile Networks [Author-notes fn1-sensors-22-06199]"

_sensors, 2022, doi:10.3390/s22166199_

Round 1
Reviewer 1 Report
I enjoyed the paper topic. However, I recommend a new reading and some text editing. Several phrases are quite large and should be rewriting. English writing is lighter. In the following my comments:
Line 30: Add and “s” to propose in “This paper proposes”
Lines 58-61: You repeat “increase” 3 times in the text. Consider the alternative redaction instead: “In addition, a larger number of BS in a small area increase the number of handovers (HO) and the challenges of mobility management in B5G networks, especially for a user speed higher than 100 km/h.”
Lines 62-63: Consider this alternative text instead of the original: “… mobility management functions are essential and need continuous developments and investigations in mobile cellular networks.”
Lines 72-74: Consider this alternative text: “… mobility management. It enables the user equipment to move and change its location to a new serving cell freely and seamlessly without disconnecting.”, instead the original: “… mobility management, which enables…”
Lines 76-78: Consider this alternative text instead of the original: “Moreover, a larger number of BSs reduces the average coverage area of each BS, increasing the number of HO compared to legacy networks.”
Lines 80-82: Consider changing “Therefore” for “Thus” at the beginning of the sentence as: “Thus, developing technology that can …”, and suppressing “Thus” at the next consecutive sentences as: “Different techniques under mobility management …”
Lines 114-115: Consider suppressing “Next” and “then” and this alternative text instead of the original: “Sections 5 presents the simulation report …”
Line 122: Consider suppressing the particle “a” before “fuzzy” and this alternative text instead of the original: “… and (3) an adaptive hysteresis fuzzy inference system to …”
Lines 138-139: Consider inserting “terms of the” and this alternative text instead of the original: “the technique achieved a good HO performance in terms of the call drop ratio and HOP, it does not provide …”
Line 213: Consider modifying the sentence tense, by replacing “are” for “were” as in this alternative text: “The results were validated using MATLAB.”
Lines 265-268: The membership function is introduced as a triangle function. This function differs from earlier works in the references. The paper does not comment if the better perform of this method as described by your results is related in any way to this fact.
Lines 282-283: Consider this alternative text instead of the original: “… into four levels which can contribute to optimal performance.”
Line 298: Consider modifying “is designed” for “has been designed”.
Line 302: Consider suppressing or changing the word “crisp”. It is not clear what you want to say with this.
Line 312: The clause refers to equation (3), but this equation appears later in line 321. Consider relocating this equation earlier.
Line 325: Within the table that describes Algorithm 1: RHOT-FLC, at label 2, it appears a variable “t=1”. This variable has not been defined along the paper.
Line 338: You should replace the word “six” for “five”. There are just 5 scenarios.
Line 366: Table 4 includes a parameter calls “Maximum number of Traffic”. Consider replacing it with a more suitable name, such as “Maximum number of UE per cell”. This table also include the parameter “Transmit Power”. This name seems to be incomplete. It should be reviewed.
Line 376: HOP expression has been numbered as (3) when it should be (4). All the sequence of equations should be rectified in lines 390, 398, and 403.
Line 451: Consider suppressing the word ”by” or replacing it by “around” in the alternative text: “the lowest HOP, less than 3.6 %, while …”.
Lines 456-458: It seems that and adjective before “number” is required to complete the sense of the sentence as: “Therefore, a larger number of small cells are required …”. Further, consider this alternative text instead of the original: “which results in an increase of the HOP.”
Lines 459-460: Consider this alternative text instead of the original: “The reduction in HOP decreases HOF probability, enhancing HO performance.”
Lines 486-487: The sentence related to “the high level of HCPs …” is not clear. Would you please review and modify?
Lines 541 and 545: The metric referred as “overall gain” is quite confused since a gain is usually defined as the ratio between the output level respect to an input level takes as a reference. Consider changing the name of this metric.
Line 559: Consider replace de word "More" for "Additional".
Author Response
I enjoyed the paper topic. However, I recommend a new reading and some text editing. Several phrases are quite large and should be rewriting. English writing is lighter. In the following my comments:
Many thanks to the reviewer for the helpful and constructive comments. We have tried to proofread the paper and provide improvements in this aspect.
Reviewer #1: Comment #1: Line 30: Add and “s” to propose in “This paper proposes”
Author response: We updated the manuscript by adding “s” to “propose”. The relevant sentence is now:
This paper proposes a Robust Handover Optimization Technique with Fuzzy Logic Controller (RHOT-FLC).
Reviewer #1: Comment #2: Lines 58-61: You repeat “increase” 3 times in the text. Consider the alternative redaction instead: “In addition, a larger number of BS in a small area increase the number of handovers (HO) and the challenges of mobility management in B5G networks, especially for a user speed higher than 100 km/h.”
Author response: We updated the manuscript by replacing the text with the suggested text. The new text is now:
“In addition, a larger number of BS in a small area increase the number of handovers (HO) and the challenges of mobility management in B5G networks, especially for a user speed higher than 100 km/h.”
Reviewer #1: Comment #3: Lines 62-63: Consider this alternative text instead of the original: “… mobility management functions are essential and need continuous developments and investigations in mobile cellular networks.”
Author response: We updated the manuscript by replacing the original text with the suggested text. The new text is now:
“Consequently, mobility management functions are essential and need continuous developments and investigations in mobile cellular networks.”
Reviewer #1: Comment #4: Lines 72-74: Consider this alternative text: “… mobility management. It enables the user equipment to move and change its location to a new serving cell freely and seamlessly without disconnecting.”, instead the original: “… mobility management, which enables…”
Author response: We updated the manuscript by replacing and improving the original text with the suggested text. The new text is now:
“HO is one of the main functions of mobility management. In the ideal case, it enables the user equipment (UE) to move and change its connection to a new serving cell seamlessly without interruption.”
Reviewer #1: Comment #5: Lines 76-78: Consider this alternative text instead of the original: “Moreover, a larger number of BSs reduces the average coverage area of each BS, increasing the number of HO compared to legacy networks.”
Author response: We updated the manuscript by rewriting and improving the original text as suggested. The new text is now:
“The use of mmWave lead to reduced cells’ coverage and a larger number of deployed small BSs thus further increased number of HO compared to legacy networks.”
Reviewer #1: Comment #6: Lines 80-82: Consider changing “Therefore” for “Thus” at the beginning of the sentence as: “Thus, developing technology that can …”, and suppressing “Thus” at the next consecutive sentences as: “Different techniques under mobility management …”
Author response: We updated the manuscript by rewriting and improving the original text according to the reviewer’s suggestion. The revised text:
“Therefore, the HO process in new mobile networks must be fast and the connection should be more seamless as many applications require very low latency and zero link failures. Thus, developing a handover technique that can provide a seamless and robust HO process is a critical challenge that needs to be solved in B5G networks. Different solutions have been proposed and developed to solve mobility issues in 5G and B5G networks [4-7].”
Reviewer #1: Comment #7: Lines 114-115: Consider suppressing “Next” and “then” and this alternative text instead of the original: “Sections 5 presents the simulation report …”
Author response: We updated the manuscript by rewriting the relevant paragraph according to the reviewer’s suggestion. The new paragraph is now:
“The rest of the paper is structured as follows: Section 2 presents recent related works, while Section 3 introduces the proposed technique (RHOT-FLC) and system model. Section 4 describes the simulation environment and performance metrics, then Section 5 presents the simulation report and performance analyses. Finally, Section 6 concludes our paper.”
Reviewer #1: Comment #8: Line 122: Consider suppressing the particle “a” before “fuzzy” and this alternative text instead of the original: “… and (3) an adaptive hysteresis fuzzy inference system to …”
Author response: We updated the manuscript by suppressing the “a”, as suggested. We also, replaced the original text with the suggested text. The new updated text is now:
“(1) AHP for defining criteria weights, (2) TOPSIS for ranking the chosen destination cells, and (3) an adaptive hysteresis fuzzy inference system to perform the calculation using parameters with direct impact on HO.”
Reviewer #1: Comment #9: Lines 138-139: Consider inserting “terms of the” and this alternative text instead of the original: “the technique achieved a good HO performance in terms of the call drop ratio and HOP, it does not provide …”
Author response: We updated the manuscript by rewriting the original text according to the reviewer’s suggestion. The new text is now:
“Although the technique achieved a good HO performance in the call drop ratio and HOP, it does not provide exemplary performance in the other HO's KPIs, such as RLF and HOPP.”
Reviewer #1: Comment #10: Line 213: Consider modifying the sentence tense, by replacing “are” for “were” as in this alternative text: “The results were validated using MATLAB.”
Author response: We updated the manuscript by changing the sentence tense from the present tense to the past tense as suggested. The new sentence is now:
“The results were validated using MATLAB Software.”
Reviewer #1: Comment #11: Lines 265-268: The membership function is introduced as a triangle function. This function differs from earlier works in the references. The paper does not comment if the better perform of this method as described by your results is related in any way to this fact.
Author response: We updated the manuscript by adding further explanation that describes the relevance of FLC triangle function and the proposed method. The updated paragraph:
“FLC has been widely applied to mobile network parameter optimization as in studies by [14,15,22,26-31] for HO. FLC has the advantages of working with imprecise inputs, not needing an accurate mathematical model, and handling nonlinearity, as proved by the results of this research.”
Reviewer #1: Comment #12: Lines 282-283: Consider this alternative text instead of the original: “… into four levels which can contribute to optimal performance.”
Author response: We updated the manuscript by replacing the original text with the suggested text. The new text is now:
“Both TTT and HOM are divided into four categories which can contribute to enhancing HO performance.”
Reviewer #1: Comment #13: Line 298: Consider modifying “is designed” for “has been designed”.
Author response: We updated the manuscript by modifying the tense of the sentence, from the past tense to the past perfect tense. The modified sentence is:
“Moreover, the proposed system has been designed to automatically adjust both TTT and HOM concurrently, unlike many works which focus on adjusting only one HCP, either TTT or HOM [8],[15],[16].”
Reviewer#1: Comment #14: Line 302: Consider suppressing or changing the word “crisp”. It is not clear what you want to say with this.
Author response: We updated the manuscript by suppressing the word “crisp”. The new text is now:
“The first stage is the input stage, which consists of three measured values as updated input from Algorithm 1 (UE’s velocity, RSRP, and RSRQ).”
Reviewer #1: Comment #15: Line 312: The clause refers to equation (3), but this equation appears later in line 321. Consider relocating this equation earlier.
Author response: We updated the manuscript by moving up equation (3), (now it is equation (2)), following the corresponding text. The text, clause, and equation (2) are now as:
“1. The RSRP for all gNBs is sorted and compared to the gNBs target station. If the equation (3) condition is not fulfilled, the HO decision is not performed. Else;
Reviewer #1: Comment #16: Line 325: Within the table that describes Algorithm 1: RHOT-FLC, at label 2, it appears a variable “t=1”. This variable has not been defined along the paper.
Author response: We updated the manuscript by adding the definition to the variable “t=1” in the second line of Algorithm 1. The definition is:
“2. if t = 1 (t: simulation time)”.
Reviewer #1: Comment #17: Line 338: You should replace the word “six” for “five”. There are just 5 scenarios.
Author response: We updated the manuscript by correcting the number of the considered scenarios to become “five” not “six” as was a typo error. The relevant sentence is now:
“The UEs move in a straight way within eight directions [N, NE, E, SE, S, SW, W, and NW] within the simulation environment and pass-through BSs with five different scenario speeds, which are 20 km/h, 40 km/h, 80 km/h, 120 km/h, and 160 km/h.”
Reviewer #1: Comment #18: Line 366: Table 4 includes a parameter calls “Maximum number of Traffic”. Consider replacing it with a more suitable name, such as “Maximum number of UE per cell”. This table also include the parameter “Transmit Power”. This name seems to be incomplete. It should be reviewed.
Author response: We updated the manuscript by replacing “Maximum number of Traffic” with “Maximum number of UE per cell”. We have also replaced the word “transmit power” with “Transmission Power” to give accurate and complete meaning.
Reviewer #1: Comment #19: Line 376: HOP expression has been numbered as (3) when it should be (4). All the sequence of equations should be rectified in lines 390, 398, and 403.
Author response: We updated the manuscript by rectifying the sequences of all the equations provided in the manuscript.
Reviewer #1: Comment #20: Line 451: Consider suppressing the word ”by” or replacing it by “around” in the alternative text: “the lowest HOP, less than 3.6 %, while …”.
Author response: We updated the manuscript by suppressing word “by”. The new sentence is now:
“RHOT-FLC has obtained the lowest HOP of less than 3.6%, while the Conv, FLC, and Slv achieved HOP of 37%, 25.7%, and 74%, respectively.”
Reviewer #1: Comment #21: Lines 456-458: It seems that and adjective before “number” is required to complete the sense of the sentence as: “Therefore, a larger number of small cells are required …”. Further, consider this alternative text instead of the original: “which results in an increase of the HOP.”
Author response: We updated the manuscript by adding “a large” before the word “number” to make the sentence complete and accurate. Also, we have replaced the original text with the suggested text. The new text is now:
“Therefore, a large number of small cells are required to be deployed in a small area, which increases the HOP .”
Reviewer #1: Comment #22: Lines 459-460: Consider this alternative text instead of the original: “The reduction in HOP decreases HOF probability, enhancing HO performance.”
Author response: We updated the manuscript by replacing the original text with the suggested text. The new text is now:
“The reduction in HOP decreases HOF probability, enhancing HO performance.”
Reviewer #1: Comment #23: Lines 486-487: The sentence related to “the high level of HCPs …” is not clear. Would you please review and modify?
Author response: We updated the manuscript by revising and rewriting the original sentence, and broke it into 2 sentences, to give a clearer meaning. The new sentences are now:
“This phenomenon may be justified due to the inappropriate setting of HCPs. Furthermore, adjusting the HCPs with high-level values caused the high probability of the RLF.”
Reviewer #1: Comment #24: Lines 541 and 545: The metric referred as “overall gain” is quite confused since a gain is usually defined as the ratio between the output level respect to an input level takes as a reference. Consider changing the name of this metric.
Author response: We updated the manuscript by replacing the metric “overall gain” with “overall improvement” in all mentioned places to give a more accurate sense.
Reviewer #1: Comment #25: Line 559: Consider replace de word "More" for "Additional".
Author response: We updated the manuscript by replacing the word “More” with the word “Additional” as suggested. The new sentence is now:
“Additional HO performance metrics, such as RLF and frequency efficiency with higher mobile speed scenarios, will be considered in future works.”

Reviewer 2 Report
The authors of this article proposed a method based on fuzzy logic technique, called Robust Handover Optimization Technique with 30 Fuzzy Logic Controller (RHOT-FLC), to configure the handover control parameters, such as time-to trigger HO and handover margin. The authors used simulation results in a simulated environment and configuration to demonstrated that the performance of their method achieves impressive performance that the others in the literature in terms HO probability, HO failure, etc.. The article describes their algorithm well and did very good survey to other algorithms. However, the article does not have sufficient discussion the reasons that their algorithm is better than the others. In addition, the simulation setup appears have some limitation to make fare comparison. Some detailed comments below:
- P2, L60: “In B5G networks” should be “in B5G networks”
- P2, L66: It is not clear what “This” refers to.
- P2, L78: Three sentences starting from this line are structured as “Therefore, …. Therefore, …. Thus, …”. It can be written in a better way.
- P6, L229: Saying “HOM is a variable that indicates the threshold difference between…” is not accurate. It may be clearer to say that “HOM is a variable that indicates the threshold of the difference between … to trigger an HO decision.”
- P7, L270: Eq (1) and (2) are equivalent. But authors gave no explanation about the relation between them. It may be easier just deleting one of them. Also, the range of x in Eq(2) should be mentioned. In addition, although this type of member function is commonly used in FL, in the following section in which rules are described, how those rules relate to this function should also be described.
- P7, L275: This section describes the proposed algorithm. It has some details, but more may be needed. For example, the authors mentioned that “Both TTT and HOM are divided into four levels which can contribute to giving optimum performance.” Then, why does four levels contribute to optimum performance? Can three levels or five levels do the same? Also, optimum in what sense should also be provided.
- P11, L335: (3000x3000) m^2?
- P11, L336: It says “the distance between each BS is 200 m”. What does it mean by “between each BS”? Shouldn’t it be “between two BSs”? If so, the figure below shows the distance between BS is 400m. Which one is correct? Please clarify.
- P11, L341: It says “until it reaches the edge of the defined area and changes its movement direction randomly within the eight possible directions”. Is it possible? For example, if an UE moves from W to E and hits the edge, how can it move E, which is one of the eight possible directions, further? Also, in 3GPP, most of system simulations set the area using wrap around method to avoid “edge effect”. It may be easier to test your algorithms with such a setup.
- P11, L350: Since only 10 UEs’ HO performance is measured, It is not clear why the simulation needs to drop 200 UEs per cell. In other words, how does 200 UEs data traffic impact on the HO performance of 10 UEs? Please clarify.
- P12, L367: It would be better to define those metrics when they are first introduced in the previous section, rather in section 4.
- P12, L376: Eq. (3) shows each UE may have different probability of HO. Why is that? Please clarify.
- P13, L404: It would be clearer to say “? is the corresponding index of the measured user” Also, it would be better to explain why different user may have different HOPP.
- P13, L425: In this section, the authors demonstrated supreme performance of proposed RHOT-FLC over the others. However, there is lack of analysis and explanation the reasons why there are such a bit differences. Also, saying that the proposed algorithm provides optimal HO performance, mentioned in several places in this section, may be too extreme since there is really no analysis to prove it in this article, but just simulation results.
Author Response
The authors of this article proposed a method based on fuzzy logic technique, called Robust Handover Optimization Technique with 30 Fuzzy Logic Controller (RHOT-FLC), to configure the handover control parameters, such as time-to trigger HO and handover margin. The authors used simulation results in a simulated environment and configuration to demonstrate that the performance of their method achieves impressive performance that the others in the literature in terms HO probability, HO failure, etc.. The article describes their algorithm well and did a very good survey of other algorithms. However, the article does not have a sufficient discussion of the reasons that their algorithm is better than the others. In addition, the simulation setup appears to have some limitations in making a fair comparison. Some detailed comments below:
Many thanks to the reviewer for the helpful and constructive comments.
Reviewer #2: Comment #1: P2, L60: “In B5G networks” should be “in B5G networks”
Author response: We updated the manuscript by correcting the capital letter in “In” with a small letter “i”.
Reviewer #2: Comment #2: P2, L66: It is not clear what “This” refers to.
Author response: We updated the manuscript by improving and revising the original text to be clearer and more understandable. The new text is now:
“Consequently, mobility management functions are essential and need continuous developments and investigations in mobile cellular networks. That involves various mobility management functions, such as the self-optimization for HO control parameters, handover decision algorithms, mobile routing algorithms, authentications, identification and tracking of changes in user location connected to the cellular network. Furthermore, mobility management provides network connectivity to users at any location. Users can avail of this function to access the network at a new location smoothly. Also, it ensures users with an uninterrupted and reliable connection, communication, and service [2]. In B5G systems, the importance of mobility management is significantly increased because many applications are very connectivity sensitive to the networks [3].”
Reviewer #2: Comment #3: P2, L78: Three sentences starting from this line are structured as “Therefore, …. Therefore, …. Thus, …”. It can be written in a better way.
Author response: We updated the manuscript by improving the sentences to be more understandable and clearer. The new paragraph is now:
“HO is one of the main functions of mobility management. In the ideal case, it enables the user equipment (UE) to move and change its connection to a new serving cell seamlessly without interruption. In B5G networks, HO is becoming more challenging, especially with high-frequency bands (such as mmWave) and ultra-dense small cells. The use of mmWave leads to reduced cells’ coverage and a larger number of deployed small BSs, thus further increasing the number of HO compared to legacy networks. This affects the network performance regarding signaling load, connectivity and throughput. Therefore, the HO process in a new mobile network must be fast and the connection should be more seamless as many applications require very low latency and zero link failures. Thus, developing a handover technique that can provide a seamless and robust HO process is a critical challenge that needs to be solved in B5G networks. Different solutions have been proposed and developed to solve mobility issues in 5G and B5G networks [4-7]. .”
Reviewer #2: Comment #4: P6, L229: Saying “HOM is a variable that indicates the threshold difference between…” is not accurate. It may be clearer to say that “HOM is a variable that indicates the threshold of the difference between … to trigger an HO decision.”
Author response: We updated the manuscript by improving the text according to the reviewer’s suggestion. The new text is now:
“HOM is a variable that indicates the threshold of the difference between the strength of the signal received at the serving base station and that of signals received at target BSs.”
Reviewer #2: Comment #5: P7, L270: Eq (1) and (2) are equivalent. But authors gave no explanation about the relation between them. It may be easier just deleting one of them. Also, the range of x in Eq(2) should be mentioned. In addition, although this type of member function is commonly used in FL, in the following section in which rules are described, how those rules relate to this function should also be described.
Author response: We updated the manuscript by removing Equation (2), as suggested. We also have added a further description of the used function and formulated rules. The relevant text is:
“The system consists of 48 rules based on three input parameters which are UE velocity, RSRP, and RSRQ. These rules are used to dynamically estimate two different outputs, TTT and HOM, in every single process. The 48 rules are formulated according to Table 2 and the triangle function, Eq. 1.”
Reviewer #2: Comment #6: P7, L275: This section describes the proposed algorithm. It has some details, but more may be needed. For example, the authors mentioned that “Both TTT and HOM are divided into four levels which can contribute to giving optimum performance.” Then, why does four levels contribute to optimum performance? Can three levels or five levels do the same? Also, optimum in what sense should also be provided.
Author response: We updated the manuscript by adding further details illustrating the explanation of using four levels of HCPs. The relevant text is:
“Both TTT and HOM are divided into four categories that can enhance HO performance. Increasing the number of levels may enhance the accuracy of selecting the TTT and HOM values. However, it increases the processing time and vice versa. Therefore, the four levels are selected to compromise the system performance, accuracy and processing time. In other words, the four levels maintain to achieve optimal HO performance, as presented in the Results and Discussions’ Section.”
Reviewer #2: Comment #7: P11, L335: (3000x3000) m^2?
Author response: We updated the manuscript by correcting the unit of the environment simulation area to be in “square meter (m2)”. The corrected area is:
“ .”
Reviewer #2: Comment #8: P11, L336: It says “the distance between each BS is 200 m”. What does it mean by “between each BS”? Shouldn’t it be “between two BSs”? If so, the figure below shows the distance between BS is 400m. Which one is correct? Please clarify.
Author response: In the revised version, we have updated the section correcting the typo error. The corrected sentence is now:
The network layout consists of sixty-one gNBs with three sectors for each cell that are deployed in a simulation environment, and the distance between two BSs is 400 m (each BS covers 200 m).
Reviewer #2: Comment #9: P11, L341: It says “until it reaches the edge of the defined area and changes its movement direction randomly within the eight possible directions”. Is it possible? For example, if an UE moves from W to E and hits the edge, how can it move E, which is one of the eight possible directions, further? Also, in 3GPP, most of system simulations set the area using wrap around method to avoid “edge effect”. It may be easier to test your algorithms with such a setup.
Author response: In the simulation setup, the consideration of the network deployment scenario is as follows:
The UE starts the movement randomly in the first simulation run with eight directions and continues until they reach the end of the green bounded circle. Once a UE reaches the edge of the green area, it randomly changes its movement direction within the two nearby possible directions that are valid to be inside the green bounded circle, or back reverse from the same previous direction. This selection is performed randomly. So, as is shown in the following original Figure (1), there is an additional area beyond the green circle (around +1 cell) to avoid the edge effect and coverage failure and to make the scenario more realistic.
The illustration presented in the manuscript is to zoom in on the scenario. It enables the readers to imagine the scenario deployed in real life, which is very close to the simulation environment.
Figure 1: the simulation deployment scenario (B5G network) which shows the possible paths and directions for one UE (as an example). Large number of small cells in a small area. The green area is the defined area for the UEs to move inside.
Reviewer #2: Comment #10: P11, L350: Since only 10 UEs’ HO performance is measured, It is not clear why the simulation needs to drop 200 UEs per cell. In other words, how does 200 UEs data traffic impact on the HO performance of 10 UEs? Please clarify.
Author response: We updated the manuscript by adding a further explanation of the impact of the UEs’ traffic. The updated text is:
"In this study, a 200 UEs traffic per simulated cell has been proposed to be generated and distributed randomly throughout the coverage area. Then it is changed dynamically and randomly in each simulation cycle. This assumption is considered to simulate a real network scenario as high traffic negatively affects the overall system performance. Once the cell traffic is increased, the HOP is increased to balance cell loads. This may lead to high degradation in the network performance in terms of HOPP, RLF, interruption time, throughput and spectral efficiency. Ten UEs were chosen to be measured in this study to investigate the HO performance for different KPIs compared with the competitive methods (like the driving test in real life, where only one or two users are used to evaluate the network). Thus, when the ten UEs move within the cells, their performance will be impacted negatively or positively based on the cell loads of each serving cell.“
Reviewer #2: Comment #11: P12, L367: It would be better to define those metrics when they are first introduced in the previous section, rather in section 4.
Author response: The first mentions of the discussed KPIs are in the introduction and several times in the manuscript with no further explanations. So we have introduced them in detail in a separate subsection to enable the readers to understand them better, including their mathematical equations.
Reviewer #2: Comment #12: P12, L376: Eq. (3) shows each UE may have different probability of HO. Why is that? Please clarify.
Author response: We updated the manuscript by adding further explanations that clarify the reasons of why each UE may have a different probability. The newly updated text is:
“Each UE is moving in a random direction which means that each UE has a different location, different signal strength, and different BS coverage. Therefore, each UE may have a different HOP.”
Reviewer #2: Comment #13: P13, L404: It would be clearer to say “? is the corresponding index of the measured user” Also, it would be better to explain why different user may have different HOPP.
Author response: We updated the manuscript by replacing the sentence word “number” with “index” in the sentence of “ is the corresponding number of the measured user” as suggested. We also, added further explanations that clarify the reasons of each UE may have a different HOPP. The updated text is now:
“where, is the corresponding index of the measured user, and is the total number of measured UEs. Moreover, each UE moves in a random direction which means that each UE has a different location, different signal strength, and different BS coverage. Therefore, each UE may have a different HOPP.”
Reviewer #2: Comment #14: P13, L425: In this section, the authors demonstrated supreme performance of proposed RHOT-FLC over the others. However, there is lack of analysis and explanation the reasons why there are such a bit differences. Also, saying that the proposed algorithm provides optimal HO performance, mentioned in several places in this section, may be too extreme since there is really no analysis to prove it in this article, but just simulation results.
Author response: We updated the manuscript by providing further explanation of the reasons this technique outperforms the competitive algorithms to the relevant paragraph (second last paragraph in section 5.5). The updated relevant paragraph is now:
In summary, as the results demonstrated, RHOT-FLC outperforms all the competitive algorithms in all considered HO performance metrics, HOP, HOF, HOPP, HOL, and HIT. This explains that the proposed technique can adjust the HCPs (TTT and HOM) efficiently and appropriately. Furthermore, adjusting both TTT and HOM properly at the same time leads to substantially reducing the HOP, HOF, HOPP, HOL, and HIT, thus greatly enhancing the HO performance.
In terms of optimality, we mentioned the desire to provide optimal results only once for our work. The rest are in the related works which provide optimization test.

Reviewer 3 Report
This paper proposed a robust handover optimization technique with fuzzy logic controller, which is an interest topic in 5G. The reviewer has some major concerns as follow:
1) The author should highlight the main contributions compared with the prior works in handover management, which is a typical issue in mobile communications.
2) For the proposed algorithm, please further add the complexity analysis.
3) For the benchmark, please illustrate the reason to choose. Does this fairness is considered in comparation? E.g., with the same scenario and system parameters?
Author Response
This paper proposed a robust handover optimization technique with fuzzy logic controller, which is an interest topic in 5G. The reviewer has some major concerns as follow:
Many thanks to the reviewer for the helpful and constructive comments.
Reviewer #3: Comment #1: The author should highlight the main contributions compared with the prior works in handover management, which is a typical issue in mobile communications.
Author response: We updated the manuscript by highlighting the research contribution in several points, in the second last paragraph of the introduction section. The updated paragraph is now:
This paper extends our previous work [13], which focused on HOP and HOPP effect reductions. In general, the main contributions of this paper are summarized as in the following:
- Proposition of a Robust Handover Optimization Technique with Fuzzy Logic Controller (RHOT-FLC) to automatically adjust the HCPs more efficiently for 5G and B5G mobile networks. In other words, we developed a fuzzy-based algorithm that utilizes the advantages of the FL system to automatically adjust the TTT and HOM simultaneously. The proposed technique exploits the UE’s information such as RSRP, RSRQ, and speed to adapt the TTT and HOM as the system outputs.
- System performance evaluation in terms of HO probability (HOP), HO failure (HOF), HO ping-pong (HOPP) effect, HO latency (HOL), and HO interruption time (HIT), with different mobility speed scenarios. The RHOT-FLC technique aims to improve the HO performance in B5G mobile system in terms of these mentioned KPIs.
- Comparison and performance analysis of the RHOT-FLC technique with various techniques from the literature, such as the conventional HO method (Conv), Fuzzy Logic Controller (FLC) [14] algorithm, and another competitive algorithm Slv by Silva et al. [15].
Reviewer #3: Comment #2: For the proposed algorithm, please further add the complexity analysis.
Author response: We updated the manuscript by adding further descriptions (summary) that illustrate the complexity analysis of the competitive algorithms plus the proposed algorithm. The summary is added to the results and discussion section. The new paragraph is:
“Regarding the complexity of the competitive algorithms and RHOT-FLC, they have a slight difference. RHOT-FLC, FLC, and Slv algorithms are FL-based techniques but the RHOT-FLC technique may have higher computational complexity as a result of the technique is designed to adjust both HOM and TTT at the same time and its rules are formulated to support low and high speeds, which leads to increasing the al-gorithm’s time execution. The FLC and Slv algorithms was designed to adjust the HOM only, while the conventional method (Conv) has the lowest complexity because the algorithm basically fixes the values of the TTT and HOM at certain values based on the RSRP only; there is no optimization or adapting process. Overall, the complexity of the competitive algorithms and RHOT-FLC can be sorted from the lowest to the high-est time complexity as follows: Conv, FLC, Slv, and RHOT-FLC. Nevertheless, the RHOT-FLC significantly enhanced the HO performance as compared with the compet-itive algorithms.”
Reviewer #3: Comment #3: For the benchmark, please illustrate the reason to choose. Does this fairness is considered in comparation? E.g., with the same scenario and system parameters?
Author response: We updated the manuscript by providing further description of using the benchmark in Section 5 (second paragraph). The new updated paragraph is:
“The proposed technique is compared with three competitive algorithms, which are the techniques presented in [15] (denoted as Slv in the results and figures), conventional HO algorithm based on the quality of signal criterion plus HOM (denoted as Conv in the results and figures), and FLC [14] (denoted as FLC in the results and figures). The competitive algorithms are chosen because of their deep focus on mobility management and MRO while having similar techniques, as in FLC and Slv. Further details are presented in Table 1. Meanwhile, these three techniques have been explained and investigated in more detail as compared to the other techniques in the literature. To ensure fairness in the comparison, we used the same simulation parameters, scenario, and environment for the proposed and competitive algorithms.”

Round 2
Reviewer 3 Report
The comments had been addressed. Please consider to accept the current version.